# Tolerance in UAE Islamic Education Textbooks

**Mariam Alhashmi [1],*** , **Naved Bakali [2],*** **and Rama Baroud [1]**

[1] College of Education, Zayed University, Abu Dhabi, Al Hayer Road, Khalifah City 144534, UAE; rerobaroud@hotmail.com

[2] School of Education, American University in Dubai, Dubai, Sheikh Zayed Road, Dubai 28282, UAE

**\*** Correspondence: mariam.alhashmi@zu.ac.ae (M.A.); nbakali@aud.edu (N.B.)

**Abstract:** The concept of "tolerance" has been thoroughly promoted within educational settings in light of the increasing need to foster co-existence and to nurture peaceful climates in societies. The United Arab Emirates (UAE) Islamic education curriculum promotes the notion of tolerance as a core tenant across different grade levels. However, there is a gap in the literature investigating the approach and conceptualization of tolerance in UAE Islamic education curricula. This study employed qualitative content analysis of grade 10–12 textbooks to understand how the concept of tolerance is promoted and to identify the opportunities and gaps in teaching tolerance through these texts. The notion of tolerance in these texts was discussed through the themes of civic engagement, critical thinking, acceptance of multiplicity, justice and equity, protection from extremism, and compassion for humans. This study identified gaps in addressing tolerance and suggested other concepts that could further supplement the Islamic education program to more thoroughly address the notion of tolerance. This study argues that teaching tolerance through religious-based instruction may facilitate an exploration of effective tolerance inculcation approaches that provide insights into the field of tolerance education at large.

**Keywords:** Islamic education; textbooks; tolerance; United Arab Emirates; multicultural education; religious instruction; cultural instruction; peace education

## 1. Introduction

Among other factors, education plays a primary role in character development and instilling the values of peace, tolerance, and coexistence. Moral development is a prime goal of Islamic education and is embedded in Islamic educational praxis (Memon and Alhashmi 2018). This paper aims at investigating the potential role of Islamic education in nurturing tolerant conduct that is grounded in traditional Islamic values. The United Arab Emirates (UAE) Islamic education curriculum documents place tolerance as a core tenant (Bakali et al. 2018), but the written curriculum has not been investigated in regard to its approach towards promoting tolerance. This study examines the theme of tolerance in the UAE Islamic education textbooks in the senior secondary levels (grades 10–12) to understand the approach through which the curriculum inculcates this value. The two primary goals for this research study are: (1) to understand how the concept of "tolerance" is conceptualized through the Islamic education textbooks in the UAE; and (2) to understand the opportunities and the gaps in teaching tolerance through Ministry of Education (MoE) sanctioned Islamic education textbooks. The UAE is home to more than 200 nationalities with differing ethnicities, religious traditions, and cultural practices (Khan 2016). As the UAE is a nation that is composed of such a multiplicity of cultures, ethnicities, and religions, the promotion of tolerance as a societal value is essential for social cohesion and coexistence.

While tolerance is not a novel concept, it is of utmost importance at present with the proliferation of intolerant and extremist viewpoints globally. The triliteral root of the Arabic term for tolerance,

*Tasamuh* or *Samaha,* is *smh* and refers to that which is easy, free from knots, easygoing, and generous (Majma'al-Lughah al-'Arabiyah 1960). The linguistic meaning of this term at its essence is good character and 'an easygoing attitude toward life and others that does not allow for harshness, intolerance, or fanaticism' (Yusuf 2005). The dimensions of tolerance in an Islamic sense require an investigation into the semantic field of this word. Yusuf (2005) notes that tolerance as *samaha* is related to the words *tashil* (to smoothen), *taysir* (facilitation), *jud* (generosity), *sakka'* (facilitating the needs of those seeking help), *karam* (generosity and nobility), *lin* (easiness, softness, and gentility), and *hilm* (forbearance). Forbearance here refers to the generosity of spirit and the ability to bear others' shortcomings out of compassion. *Samaha* also connotes vastness as it means "to permit". This linguistic deconstruction of terms surrounding the notion of tolerance within an Islamic world view has led some scholars to describe the faith as *al-hanafiyya al-samha,* which means a religion without any constriction or rigidity in it (Yusuf 2005). This does not preclude the existence of other interpretations. Some Muslim groups have had more confrontational and constricted views of Islam, particularly in relation to other religious traditions. These groups view Islam and Muslims as diametrically opposed to an existential non-Muslim 'other' (Maher 2016). It has been argued that these confrontational views are rooted in theology and thus explain the proliferation of Islamic terrorist organizations such as al-Qaeda and ISIS (Maher 2016). However, this is an oversimplified explanation of these groups and participation within them. Rather, the growth of terrorist organizations that claim to act in the name of Islam is a complex multifaceted phenomenon influenced by geopolitical realities, economic and political instability, and other social factors (Sageman 2016). The analysis, which follows, situates tolerance within an Islamic worldview as encompassing the qualities of generosity, softheartedness, and compassion (Yusuf 2005).

Tolerance can be explained through a broad spectrum of definitions. The difference in definitions is attributed to the difference of cultures and to the variety of epistemological frameworks that are used to construct or understand the concept. Western frameworks build on sociological theories, social interaction theory, conflict theory, frustration-aggression theory, and the theory of emotional culture (Boghian 2017). From this perspective, tolerance can be defined as "respect, acceptance, and appreciation of the richness and diversity of our world's cultures, ways of speaking and expressing our quality of human beings" (Boghian 2016). Tolerance, according to UN General Assembly Resolution No. 51/201, is defined as "respect, acceptance and appreciation of the endless richness of our world's cultures, our forms of expression and ways of being human" (Member States of UNESCO 1996). Shaykh Abdullah Bin Bayyah, recognized by Muslim scholars as one of the greatest living authorities on Islamic legal methodology, has spoken at length on Islamic conceptions of tolerance (The Official Portal of the UAE Government 2020, 15 May). He stated in his keynote speech at the Forum for Promoting Peace in Muslim Societies (Bin Bayyah 2019a) that religion is akin to energy in that it can facilitate flourishing and stability or wreak havoc and destruction. He proposed that the essential teachings [of Abrahamic religions] concerning peace, coexistence, universal human dignity, and respect of religious differences can work as powerful antidotes against violent religious extremism. The Islamic discourse is characterized by its vastness in accommodating diverse points of view and opinions and embracing disagreement. Yusuf (2005) illustrated how all five golden principles of Islamic jurisprudence relate ultimately to the concept of *samaha*. These five principles are: 'Affairs are determined by their ends and aims', 'Harm must be removed', 'Customs are afforded legal status', 'Certainty is not removed by doubt', and 'Difficulties demand facilitation'. Daud (2014) discussed the motivations for tolerance within an Islamic framework. He suggested that there are three motivations for tolerating things that one might not like. The first is due to powerlessness and the second is due to indifference. He argues that the third is implied by the Islamic understanding of tolerance, which is tolerance out of love and humility. According to Bin Bayyah (2019b), tolerance in Islam builds an integrated culture that has its unique values, manifestations, and fields as well as methodological foundations that it is based on. These three foundations include: the Islamic outlook towards the other where Islam considers people as brothers; the Islamic position towards difference and diversity in which Islam recognizes difference

as a universal principle; and the place of dialogue in Islam where it is a religious duty and a necessity for humanity.

This study defines tolerance, according to the Islamic methodological framework outlined above, as dealing with the 'other' out of mercy and ease and confronting the offender with forbearance, compassion, and generosity, based on understandings to promote justice (*'adl*).

When broadly exploring the notion of tolerance in Western contexts, questions inevitably arise regarding the extent to which tolerance trumps freedom of expression, when should freedom of expression supersede tolerance, and if it is possible to tolerate one thing without discriminating against another. The notion of tolerance gained prominence throughout the Enlightenment, as there was a growing emphasis on individual autonomy and arbitration for competing truth claims during this period. Furthermore, emphasis on tolerance was perceived to be a consequence of the allegedly religious nature of the wars during the post-reformation period (Horton and Mendus 1985). As such, the notion of tolerance, as developed through the Western experience, was intended to assuage tensions at a time when there were competing views on politics and religion flourishing in European society, to facilitate co-existence and social cohesion. Given the potential sources of tension that may arise through the promotion of tolerance, it is important to understand that tolerance, as framed through Western liberal discourse, may not necessarily be the standard view or understanding of tolerance that is or should be promoted in non-Western educational contexts. Tolerance as a Western construct is often synonymous to acceptance. It refers to respecting the freedom of the other, their way of thinking, their behaviors, and their political and religious opinions (Bin Bayyah 2019a). Aristotle describes a virtue as a "mean" or "intermediate" between two extremes. On that principle, Daud (2014) points to the fact that tolerance too has limits that should be determined by the experts in each field. Tolerance in the Islamic tradition does not imply a relativity of truth. Al-Jifry (2018) discussed the limits of tolerance in his analysis of the roots of tolerance in the Islamic tradition based on the works of Imam al-Ghazali (d. 1111). He stated that tolerance, according to al-Ghazali, 'is not the misguided tolerance that strips *haq* (truth) of its innate value or that which establishes for indifference and contentment with the rejection of truth nor the satisfaction with ignorance. Rather, it is the type of tolerance that is in line with *'aql* (mind) and *'adl* (justice)' (p. 14). This is the notion of tolerance that is considered a key component of education in the Islamic tradition.

Religious education is often found to promote tolerance (e.g., Barnes 1997; Saeed 1999; Schweitzer 2007; Abu-Nimer et al. 2016; Afrianty 2012; Vanner et al. 2017; Waghid and Davids 2014; Jones 2007; Kyl 2004; Baligadoo 2014; Naumenko and Naumenko 2016; Fujiwara 2007; Halai and Durrani 2018; Das (Sarma) (2013); McEvoy-Levy 2017; Rashed 2015). Internationally, the role of Islamic schools has been debated in relation to their possible influence on religious intolerance (Elbih 2012). For example, Islamic education in the context of Pakistan has fueled intolerant religious discourse in the nation (Talbani 1996). Hundreds of religious schools and programs worldwide promote the spreading of tolerance and co-existence. One such initiative that claims achieving this objective is the Imam-Hatip Islamic schools in Turkey (Alamac and Kaymakcan 2017). Schools in Singapore have also attempted to promote religious harmony through actively promoting the values of peace and tolerance in their Civics and Moral Education curricula (Tan 2008). Furthermore, the themes of religious tolerance have been investigated in Islamic education high schools in Indonesia (Wekke et al. 2017) and the conceptualization of peace education in these schools was researched (Taufik 2016). However, there is a gap in the literature with regards to how Islamic education curricula in the UAE aim to promote tolerance and peaceful coexistence.

The UAE government declared that its Constitution incorporates values of tolerance such as equality, freedom, and respect (The Official Portal of the UAE Government 2019). The UAE appointed its first Minister of State for Tolerance in February 2016 to contribute to the country's efforts towards consolidating the values of tolerance. A National Program of Tolerance was launched in 2016 to promote the values of tolerance and ensure their future sustainability. Tolerance in the UAE is framed around the concept of "Middleness" (moderation) which is derived from the Islamic tradition. Moderation

refers to 'just and temperate behavior in various spheres of life' (Afsaruddin 2009, p. 352). In 2016, the program of tolerance mentioned that the Ministry of Tolerance bases the values of tolerance on seven pillars that include: Islam, the UAE constitution, Shaykh Zayed's legacy (the founder of the country), international conventions, archeology and history, human nature, and common values (The Official Portal of the UAE Government 2020). To further promote tolerance, the year of 2019 was announced by the President of the UAE as the Year of Tolerance. As part of the UAE initiatives in implementing the concept of tolerance in the curricula, the Ministry of Education launched several initiatives such as that of 'communication of civilization' to contribute to the teaching of tolerance. Tolerance was reinforced amongst students from different communities in the country through traditional practices of hospitality, and respect for the elders. Having described the notion of tolerance from various cultural, historical, and religious perspectives, the paper now turns to explain the methodological process employed to investigate tolerance in senior Islamic Education textbooks used in the UAE.

## 2. Methodology

A textual analysis of the Islamic education textbooks was utilized through directed qualitative content analysis (Potter and Levine-Donnerstein 1999; Mayring 2000; Hsieh and Shannon 2005). The Islamic education textbooks in the UAE are centrally developed by the Ministry of Education and are provided to all public and private schools (Bakali et al. 2018). The high school Islamic education textbooks for the academic year 2018–2019 for grades 10–12 were selected because the curriculum in these grade levels places a focus on inculcating an awareness of pressing issues related to social cohesion, differences in culture, and globalization. Additionally, evidence suggest that youth in this age range are more prone to be targeted by violent extremist organizations (Al-Rabaani 2018). Hence, instilling tolerance becomes an issue of vital importance at this stage. The process of textual analysis took place through the following five phases:

1. *Category Development.* An initial list of categories was generated by drawing from previous studies and theories in the fields of Islamic education and tolerance, multicultural and diversity education (Potter and Levine-Donnerstein 1999; Stemler 2001). The researchers worked independently to develop categories to investigate the theme of tolerance and religious education contained in the textbooks (Neuendorf 2002). This approach entailed using existing research findings and theories to formulate the initial categories (Hsieh and Shannon 2005) through deductive category development. The categories formed benchmarks for the researchers to determine how the overarching theme of "tolerance" appeared in the textbooks. To ensure the validity of the selected categories, we consulted with professors and other experts in fields related to education, tolerance, and religious instruction to ensure that the chosen categories accurately represented the concept of tolerance for the purposes of our investigation. Next, we unified the list of categories and developed a manual. In order for the content analysis to be efficient, the categories were operationally and clearly defined at the outset (Guthrie et al. 2004). The manual also included descriptions in order to determine exactly under what circumstances units of data would be placed under a given category. As we analyzed more data, new concepts and themes emerged, which were added to the manual.

2. *Phase Two: Coding System Testing.* In the second phase, we tested the coding system through categorizing sample text, checking consistency among the researchers, and revising rules until sufficient consistency was achieved (Weber 1990). According to Markus and Adler (1999), content analysis should demonstrate reliability for both the employed instrument and the data in order to draw valid inferences.

3. *Phase Three: Text Coding.* In the third phase, textbooks were carefully read, and portions of the text were highlighted that appeared to describe the concepts closely related to the defined categories. The highlighted text was then coded using the predetermined categories in the manual that was created in an Excel spreadsheet. The data that could not be coded under the predetermined categories were analyzed later to decide whether they described a new category or subcategory.

Developing new categories was limited, as we broadly defined the categories to accommodate for the large amount of data that was analyzed (Morse and Field 1995).

4. *Phase Four: Consistency Check After Coding the Textbooks.* The fourth phase involved rechecking the coding categories and the text analyzed for consistency to ensure that the data was coded in a reliable and consistent manner across the eight textbooks.

5. *Phase Five: Data Analysis.* In the final phase, we analyzed coded data to understand the identified categories and themes and their properties. As the qualitative content analysis did not contain statistical figures and counts, our analysis focused on exploring patterns, themes, and characteristics.

## 3. Findings

The findings section reports on the concepts of tolerance that were presented in the textbooks. These concepts include the direct discussion of the concept of tolerance, civic engagement, critical thinking, respect of diversity, justice and equity, moderation, awareness of extremism, compassion for humans, and conflict resolution. This section discusses the ways in which these concepts were addressed. The underrepresented concepts were also identified.

### 3.1. Tolerance

Dedicated lessons on the concept of tolerance appeared in both volumes of the grade 10 textbooks in a way that rooted the concept in authentic Islamic discourse. In volume one, "tolerance" was discussed through the Prophetic model of tolerance, historic examples, love for the "other", forgiveness, and the importance of tolerance in our lives. The second volume of the grade 10 textbook had an entire chapter devoted to the theme of "tolerance". The textbook stated that 'Islam urges Muslims to exercise tolerance of people in all spheres of life' (Grade 10, Part 2, p. 212). The four main components of tolerance, as conceptualized in these texts, were: mercy towards others, forgiveness of people's transgressions, kind dialogue in disagreement, and doing good to others (Grade 10, Part 2, p. 213). Furthermore, the concept of tolerance in these texts implied a rejection of violence as a means for resolving disputes and accepting ways that differ from normative Sunni Muslim practice, by not compelling others to act in accordance with one's own will. In relation to this point, the textbook referred to the "Charter of Madinah", which was a constitution established in early Madinan society, and noted that "the Charter of Madinah comprised nearly 52 articles, twenty-seven of which related to the relationship between Muslims and followers of different religions" (Grade 10, Part 2, p. 216). According to the textbook, the Charter protected individual rights for all the inhabitants of Madinah, regardless of their faith. The protection of rights as an expression of tolerance was not limited to the inhabitants of Madinah, as prisoners of war in the time of the Prophet were given priority over soldiers for food rations (Grade 10, Part 2, p. 217). Additionally, tolerance in this chapter implied promoting justice and equity for all people. The textbook observed, "One aspect of respecting others and observing their rights in full, irrespective of differences relating to colour, religion or race is exercising tolerance in our life" (Grade 10, Part 2, p. 219). This discussion is preceded by a story from the early companions of the Prophet and their conquest of Jerusalem. The textbook described how the Islamic Caliph, Omar, assured the inhabitants of the holy city that their religious beliefs and places of worship would be preserved and protected.

### 3.2. Civic Engagement

Civic engagement stems from the notion of generous contribution and facilitation that is embedded in the definition of tolerance. Civic engagement and national belonging have been carefully woven throughout the Islamic education textbooks and contribute to a well-considered approach towards presenting the lessons to students. The introduction stated that nurturing citizenship was one of the main objectives of the Islamic education curriculum. 'The book aims at inculcating the traits of the Emirati citizen and to promote his/her loyalty and belonging to their country' (Grade 10, Part 1, p. 7)

The curriculum developers clearly made an effort to include aspects that related to the shaping of national identity and to contemporary issues (Grade 11, Part 2, p. 8). Volume one of the grade 10 textbook looked at political pacts during the lifetime of the Prophet to demonstrate how these themes of tolerance were essential topics in Islamic education in the UAE.

There was specific mention of the Alliance of Fudool, which was a pact that restored the rights of the oppressed in pre-Islamic Arabia, which the Prophet had described as dear to him and that he would have supported in the advent of Islam. Furthermore, the concept of civic engagement was promoted as a way to build cohesiveness in society through actions such as the kind treatment towards one's neighbor, protecting their rights, and looking out for their interests. The chapter is replete with examples and stories of the early companions of the Prophet who served society. Furthermore, the grade 10 Islamic education textbook touched on the theme of prioritizing security and safety in society. As such, tolerance was conceived of as a way to keep peace and foster cohesiveness amongst all members in society. Societal security and safety was also alluded to through discussion of the hypocrites of Madinah. This was a group of people in Madinan society who secretly tried to undermine the fledgling Muslim community in Madinah through conspiring with enemies of the state, spreading false rumors in the community, and covertly engaging in sedition. Despite the challenges that this segment of society posed to the Muslim community in Madinah, the textbook describes how the Prophet did not harm them or name them in public, with the intention of avoiding unrest in society and to preserve social harmony. The implication expressed in the textbook was that, by exposing the hypocrites and publicly naming them, it would cause tribal hostilities, which would damage the fabric of the newly formed Islamic state. The grade 10 textbook also devoted an entire chapter on the late wife of the founder of the UAE, Sheikha Fatima bint Mubarak, and described how her efforts served as an example of civic engagement and serving one's society. The textbook described her empathy towards all citizens as a form of tolerance in UAE society, as she founded a number of charities and aid organizations for peoples with disabilities, the elderly, orphans, and other marginalized members of society of all faiths. Defending one's homeland was also described as a form of civic engagement. This was discussed in great detail through the section of the textbook which focuses on mandatory military service and described it as a way to serve one's country, promote peace and security in society, and as an essential way to serve the interests of the nation.

### 3.3. Critical Thinking

Critical thinking contributes to the nurturing of forbearance that serves as a deterrent from harshness and fanaticism. The theme of critical thinking is an important feature of the UAE Islamic education curriculum. For example, critical thinking was applied in volume one of the grade 10 textbook through criticisms of blind following in religion. The textbook warned students of the dire consequences of uncritically following and accepting religious doctrine. Any type of religious instruction, edict, or practice needs to be derived from an authentic source to prove its validity. The discussion of blind following appeared in the chapter entitled "The five purposes of legislation". Under the purpose of "preserving one's mind", one of the five purposes of divine legislation, the textbook described how using critical thinking skills and not engaging in blind following was an important component to preserving one's mind. The textbook exhorted students to free their minds from "the tyranny of myths and illusions, which are based on ignorance and blind imitation" (Grade 10, Part 1, p. 150). This theme of critical thinking, through rejecting blind following, was key to understanding the conceptual underpinnings of how "tolerance" was discussed in the UAE Islamic education textbooks. Blind following, according to the texts, could lead one into extreme and erroneous practices and beliefs, which may reinforce intolerant views.

Critical thinking was also infused as a pedagogical approach. The introduction stated that 'the educational activities [in the book] were varied to contribute to developing the learners' critical thinking, a contemporary demand that protects students from erroneous ideas and blind imitation' (Grade 10, Part 1, p. 7). The questions in the textbook often required the students to analyze the Quranic verses or

the texts of Prophetic narration and to conclude ideas and implications from these. For example, one question asked the students to discuss and analyze in groups how the independent reasoning (*ijtihad*) of the Prophet's companions in a particular situation led them to different conclusions and rulings that were all approved by Prophet Muhammad. In addition, activities of student self-assessment that require students to assess the extent to which they implement the values and principles are common in the textbooks. One example is a student self-assessment of the level of respecting the different opinions of the other and of scholars.

### 3.4. Acceptance of Multiplicity

The theme of acceptance of multiplicity comes up through discussions of how cultural diversity in the UAE has had a positive impact on society. In relation to this point, the textbook mentions, 'Nearly 206 nationalities co-exist in the United Arab Emirates, either transiting or residing; these represent more than 200 nations and use 100 dialects. These racial groups exist in complete harmony and integration. The outcome of this has had a very good effect on various spheres' (Grade 10, Part 1, p. 80). A dedicated lesson in volume one of the grade 10 textbook focused on acceptance of multiplicity of opinions. This lesson explained in detail the reasons why scholars of jurisprudence reached different opinions, the fruits of differences, and the position of the Muslims towards these differences. Another lesson discussed the values of compassion and respect between scholars of different opinions. The positions of major scholars in the Islamic tradition and how they refused to impose their opinions on people was also discussed. Furthermore, there was reference in the textbook about how Islam condemns the cursing of other religions citing Quranic verses and the hadith literature, as it could sow enmity in society and may cause those groups being attacked to revile one's own religion. Volume one of the grade 11 textbook included a lesson on the etiquette of dialogue (adab al-hiwar) and a lesson on Islam and social communication. Another example of acceptance of multiplicity in the grade 11 textbook appears in a lesson dedicated to Sukainah bint Alhusain, a highly respected figure to Shiaa' Muslims.

### 3.5. Justice and Equity

Justice (*'adl*) is framed along the theme of tolerance in volume one of the grade 10 textbook. Justice and equity were spoken of at length through examples of the Prophet's biography. There was specific mention of the "Charter of Madinah", which was a regulatory guide for peaceful co-existence between the various tribes, religious groups, and ethnicities in Madinah. The textbook described the Charter of Madinah as a set of guidelines that promoted 'sublime human principles, such as taking sides with the oppressed, protecting neighbors, taking care of private and public rights, combating crime, cooperation in paying *diyah* (blood money), freedom of creed, and helping indebted persons' (Grade 10, Part 1, p. 79). These principles were equally applied to the citizens of Madinah, regardless of their religious beliefs or tribal affiliations. As the textbook observed, 'These principles make sons of the same country, who belong to different races, ethnic origins and beliefs, feel that they constitute one family . . . Equity between all these groups was established on the basis of common humanity; people are equal in terms of the origin of human dignity' (Grade 10, Part 1, p. 79). The grade 11 textbooks touched upon tolerance through the theme of "Equity", as the textbook describes equity as "promoting tolerance" (Grade 11, Part 1, p. 157). Equity is further described as giving people their just dues regardless of their social status, religion, or culture. The grade 11 textbooks give examples linking equity with tolerance by describing verses of the Quran and setting examples from Prophet Muhammad's biography (*sirah*), which involve giving non-Muslim members of society their rights and treating them with respect and dignity (Grade 11, Part 1, p. 157).

### 3.6. Compassion for Humans

A recurring theme of how tolerance was conceptualized in the grade 10 Islamic education textbook was through showing kindness towards others who may oppose and mistreat you. This concept was

linked to examples in Prophet Muhammad's biography, which described how he showed kindness towards those in Meccan society who opposed him and mistreated him. Other formulations of tolerance made reference to stories of the companions of Prophet Muhammad, who showed kindness towards non-Muslim citizens living within the Islamic state. Compassion is also highlighted in volume two of the grade 10 textbook that narrates the story of the conquest of Makkah and how the Prophet forgave and chose to live in peace with the non-Muslim inhabitants of the city. He chose this path despite the fact that these same inhabitants just a few years earlier treated him and the early Muslim community with hostility, violence, and persecution.

*3.7. Protection from Extremism*

The textbooks consider protecting students from extremism as an aim of the curriculum: "the book aims at . . . protecting them [the students] from extreme and terrorist ideas" (Grade 10, Part 1, p. 7). A portion of the first volume of the grade 12 textbook was devoted to the topic of extremism. Extremism was discussed in a multifaceted way and emphasized that it was connected to abandoning moderation in one's approach. According to the textbooks, extremism could refer to overzealousness in religious practice and observance as well as intolerance towards differing views, both within Islam and towards other faiths. The textbooks discussed how the practice of Islam needed to adhere to a balanced approach, which was based on the example of the Prophet, who condemned individuals who engaged in extreme practices or held extreme views. Grounded knowledge and education were essential for curbing extremist views, as alluded to in the discussion of critical thinking. In the textbook, the eventual consequences of extremist views were understood to be dire and could lead to violence and extremism. As the textbook noted, "The question of intolerance and extremism is a serious question indeed; it transforms man into an enemy of his homeland, society, and relatives. . . Muslims experienced at the hands of extremists the most heinous of crimes: terrorism, murders, violation of honor, looting, diffusion of ignorance, chaos and destruction" (Grade 12, Part 1, p. 41).

Tolerance in the grade 10 textbook advocated that a lack of tolerance may put individuals on a trajectory towards radicalism and extremism. As such, there were explicit attempts to link intolerance towards violent extremism and terrorism. This trajectory towards potential violent extremism through intolerance was not clearly explained, but rather it was an underlying assumption embedded in the text.

## 4. Discussion

Tolerance education is in high demand for educational systems at an international level. This study investigated the approach of teaching tolerance through Islamic education textbooks in the UAE. It also analyzed the strengths and the gaps within the current approach. This study found that tolerance was addressed in the Islamic education curriculum through the concepts of tolerance, acceptance of multiplicity, justice and equity, civic engagement, critical thinking, compassion for human moderation, and protection from extremism. These concepts align with the underlying values of tolerance from an Islamic view, which emphasizes facilitation, generosity, and vastness. The concepts of tolerance that characterize Islamic education within the Islamic tradition including moderation and correct knowledge of religion were not discussed in sufficient detail and could be further developed. Furthermore, the concepts of gentility (*lin*) and forbearance (*hilm*) were underrepresented.

According to our interpretations, the Islamic education curriculum considers instilling tolerance and protecting students from extremism as one of its main objectives (Grade 10, Part 1, p. 7); for the textbooks to be effective, they need to discuss the theme of tolerance in a way that addresses the local push and pull factors leading to radicalization (Elsayed et al. 2017). Internet and social media (e.g., Facebook and Twitter) are the main platforms that are employed for the radicalization and recruitment of youth by extremist groups (Elsayed et al. 2017). However, these platforms and student code of conduct in virtual spaces were not addressed in the textbooks. The "How-To" guide by Elsayed et al. (2017) defines push factors as "factors that make an environment more conducive to

violent extremism" and pull factors as "psychosocial conditions that drive an individual to violent extremism" (p. 5). Amongst fifteen push and fifteen pull factors mentioned in the guide as local to the Middle East and North Africa (MENA) region, the following are related to our investigation of the UAE Islamic education curriculum:

*4.1. Push Factors*

(1) Lack of trust and attachment to State institutions and policies, particularly among youth
(2) Lack of religious awareness and religious culture due to the decline in proper religious education among religious leaders and community members
(3) Perceived ideological struggle (religious versus secular) and the perception that secular society is "corrupting" religious society
(4) Lack of critical thinking skills inculcated in schools

*4.2. Pull Factors*

(1) Provision of a sense of belonging to a group, purpose, self-realization, and fulfillment
(2) Provision of an avenue for the expression of religion (as an alternative for religious institutions)
(3) The influential role of persuasive (online) religious leaders promoting "jihad"
(4) Idealizing the narrative of dreams of living under the "caliphate" as a path to heaven through engaging in a war on perceived infidels

An examination of the findings against the abovementioned push and pull factors reveals that the textbooks address some of the push factors through proactive strategies. One of the major areas of strength is the curriculum's focus on citizenship and belonging. This is consistent with the seventh recommendation from the Global Counterterrorism Forum (GCTF) (2015): '*Increase and expand on curricula that emphasize civic education, civic responsibility and human value*'. This recommendation suggests that civic education provides youth with a framework for a collective civic identity and fosters tolerance, along with the willingness to negotiate and compromise. According to this recommendation, to be most effective, civic education and its related values must be relevant to the local context and culture which was evident in the UAE Islamic education textbooks.

There remain opportunities for making improvements and further developing the textbook discourse on tolerance. For example, critical thinking skills can be further developed through the inclusion of open-ended and unstructured questions. There is a tendency in the textbooks towards the overuse of close-ended questions that require one predetermined answer. According to the sixth recommendation from the GCTF (2015), effective strategies for countering extremist views and ideas involve developing problem-solving and critical-thinking skills. As stated: "*Emphasize in curricula the concepts of problem-solving and the examination of issues through a "gray" lens as opposed to a black-and-white lens*". Such critical-thinking skills are important for challenging violent extremist messaging, as they cause individuals to question the truth claims of such messaging and their authoritative basis. Therefore, critical-thinking skills need to be grounded and structured around a coherent knowledge framework.

In relation to framing critical thinking around an informed knowledge structure, a second opportunity pertains to building awareness of sound Islamic knowledge, which needs to be emphasized through the establishment of authoritative sources of knowledge. For example, activities in volume one of the grade 10 textbook required the students to interpret verses from the Quran in their own words. We suggest, in addition to individual contemplation, referral to authoritative resources, the methods by which these are selected, and acknowledging the difference of interpretations within certain boundaries. As such, there is a need for individual voices of the students to emerge; however, these individual expressions need to be informed opinions that have an authoritative basis. The misconception of freedom of individual interpretation can also occur as a result of the gap in volume one of the grade 11 textbook, where a lesson discussed *Shari'ah* (Islamic religious law) sources but does not address the requirements for those who are eligible to conclude rulings. In their systematic review of scientific

literature on radicalization into violent extremism, Vergani et al. (2018) have found that the most cited push factor is the justification of violence and that it is achieved by religious fundamentalists through the rejection of the mainstream religious scholarly tradition in the name of going "back to scripture". Therefore, it is important to teach students to seek out traditional and intellectually rigorous interpretations as they study the scripture. Moreover, religious literacy is also essential for students to develop a moral stance that equips them to recognize the "other" (Ghosh et al. 2017). This is needed at an inter-religious as well as an intrareligious level. Teaching a moderate understanding of Islam also contributes to the inculcation of tolerance. Vergani et al. (2018) have found that 'knowledge of Islam and religiosity are often negatively associated with radicalization' (p. 10). This was also concluded by Davies (2018), who authored a report on international educational initiatives that effectively work in counterextremism. He argued that ideological interventions that teach the "correct" versions of Islam and provide a sound theological framework have the potential to prevent violent religious extremism. A similar finding was concluded by Shakeel and Wolf (2017), who indicated that the vast majority of Islamic and reactionary American terrorists attended traditional public schools and had no religious education.

A third area that could be further developed is discussions relating to extremism in the textbooks. When extremism was discussed in the first volume of the grade 12 textbook, for example, it was done at a shallow level and not in a meaningful fashion. Extremism was not fully fleshed out to mean anything beyond religious extremism and its eventual trajectory towards terrorism. Extremist views can be violent or they can be quietest; however, there was no differentiation in the textbook. Furthermore, there was a tendency in the textbook to immediately link intolerance to extremism. The textbook did not discuss that there were different levels of intolerance and that intolerance leading to extremism was generally an exceptional situation and not a normative reaction. The UAE Islamic education textbook approach to addressing extremism was underdeveloped as it simply presumed that intolerant discourse could serve as a catalyst towards adopting extremist points of views and potentially engage in violent extremist activity. This was problematic, because there was no clear discussion of how social, economic, or political factors were drivers towards engaging in violent extremist activities. As such, discussions in the textbooks about extremism and violent extremist activity did not engage in a nuanced discussion of the phenomenon. Rather, these issues were addressed with some superficiality and presumed that discussions about tolerance and the dangers of intolerance addressed the issue sufficiently.

## 5. Conclusions

The UAE Islamic education curriculum places a dedicated focus on tolerance in multiple ways throughout the curriculum and manifests this concept from an Islamic point of view. Based on our analysis of the textbooks, we propose a tolerance education taxonomy that is organized under three categories: teaching about tolerance, teaching for tolerance, and teaching through a pedagogy of tolerance. Teaching about tolerance refers to the dedicated topics that represent the theme of tolerance. Teaching for tolerance refers to the topics that are related to tolerance and support nurturing tolerance amongst students. Teaching through a pedagogy of tolerance refers to the pedagogical approaches adopted by the textbooks that foster tolerance. In terms of "Teaching about tolerance", the textbooks discussed the theme of tolerance explicitly and adequately as described in the results. The textbooks' developers made an effort to achieve "Teaching for tolerance" by discussing the concepts of acceptance of multiplicity, justice and equity, civic engagement, critical thinking, compassion for humans, and protection from extremism.

Teaching tolerance through religious-based instruction may provide effective practices that provide insights into the field of tolerance education at large. There is an opportunity to improve on the category of "teaching through a pedagogy of tolerance" in the analyzed curriculum by developing the infusion of critical-thinking skills and awareness on discerning authentic knowledge from ideas that lack a sound scholarly basis. Notably, we cannot study textbooks in isolation from the roles of teachers as well as the guidance and the training they receive. Future studies will be needed in the

area of comparing the taught curriculum (what the teachers teach in the classroom) and the learnt curriculum (the knowledge that the students gain) to the aims and the methods of tolerance presented in the textbooks to measure their actual impact. Further research is also required to investigate the ways in which the concepts of tolerance presented in Islamic education curricula are complemented by the concepts of tolerance presented in other subject areas of the UAE curriculum such as languages, social studies, and moral education.

**Author Contributions:** M.A. was the primary researcher and writer, N.B. was involved in writing and editing, R.B. assisted with writing and editing. All authors have read and agreed to the published version of the manuscript.

**Funding:** This study was funded by Sheikh Saud bin Saqr Al Qasimi Foundation for Policy Research.

**Conflicts of Interest:** The authors declare no conflict of interests.

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
