# Peer review of "Tolerance in UAE Islamic Education Textbooks"

_religions, doi:10.3390/rel11080377_

Round 1

Reviewer 1 Report

This is a well-researched study which demonstrates methodological rigorousness. There are, however, some problems regarding the presentation of analysis and conclusions, and how these are reached. These problems make the text unbiased and less transparent.

Line 50-51 presents a conclusion about what is an Islamic point of view of a tolerant person. I suggest the authors should demonstrate care in how the conclusion is reached and presented. First of all, the conclusion is based on the previous sentences, lines 38 – 48, which address a semantic analysis of the Arabic root smh.  Based on the semantic analysis of the Arabic term, the following lines 48 – 50, then present a “description” of Islam as not being rigid. There is here a conflation between the semantic meanings of an Arabic term and some Islamic interpretation of those terms regarding what a tolerant person is. There is a need for clarification as to who describes (or interpret) Islam as al-hanfiyya al-samha. Is this the author’s interpretation, or some Islamic groups or scholars?

Lines 51- 72 presents various interpretations of Islam as being tolerant, and tolerance in Islam. This section should instead be included in lines 73 – 110, under which is discussed “definitions and meanings”. The way the text is now organised prioritizes certain Muslim voices (f. ex. Shaykh Abdullah Bin Bayyad) and gives them the authority to define the concept tolerance. Instead, the views should be contextualised as part of a larger debate.

From line 366 I suggest adding at the beginning of the sentence, “I argue/ we argue” to indicate that what follows is the author’s suggestions.

Lines 410-437. This passage raises some interesting and important issues regarding the relationship and borders between critical thinking and individual interpretation on the one hand, and authoritative sources of knowledge and scholarly tradition on the other. The author seems to be wanting both, although underlining the importance of the latter. The topic relates to the relationship between critical thinking and the construction of normative. The author seems to argue for the possibility of pupils’ critical thinking, but within certain given frames. This raises new questions about authority and legitimacy, and the impact of societal development and conditions to the evolving interpretation of Islam. I would suggest a discussion on this, perhaps with reference to how this is solved in various Islamic educations institutions and programmes.

Some editorial suggestions:

Lines 72-72 and 74-75: the causal relationship between the two sentences is unclear.

Lines 467-468 seems to be misplaced. Disrupts the flow of the discussion about teaching tolerance about, for and through religious based instruction.

Author Response

Point one: This issue has been addressed by elaborating on the discussion of tolerance and differentiating between linguistic deconstructions of the Arabic term for tolerance and how tolerance has been understood by Muslims. There was an acknowledgement that there are differing views of tolerance or its promotion amongst Muslims. Specifically, this discussion touched on the growth of Muslims that commit acts of violence and terrorism in the name of Islam and how this seemingly contradicts the notion of ‘tolerance’ as an Islamic principle. It was explained in this discussion that the existence of these groups does not preclude tolerance as an Islamic value, rather the existence and growth of these groups are a complex phenomenon explained by a variety of geopolitical and social factors.

Point 2: The text was rearranged according to the reviewers suggestions. A few sentences were added and deleted so that the rearrangement was coherent. This re-arrangement, as the reviewer suggests, helps situate our analysis of tolerance within a larger debate and discussion.

Point 3: To address this point, we added the ‘according to our interpretations’, which clearly demonstrates that the analysis is based on our views and understandings of the texts.

Point 4: The points brought up by the reviewer here were reconciled by discussing how ‘critical thinking’ needs to be based on an informed opinion. In other words, one can be critical and exercise critical thought, but it needs to be grounded in a knowledge framework. The added sentences in this section has better linked suggestions 1 and 2 in the discussion section to create a more coherent argument. 

Editorial suggestions:

Point 1: This point was addressed through point 2 mentioned above. By rearranging the sections and the light editing for coherence this point was resolved.

Point 2: The sentence in question was removed, as it was out of place

Reviewer 2 Report

No comments, I appreciated the work very much.

Author Response

Many thanks for your review of our manuscript. No comments to respond to for this reviewer